# Urbanisation in Sub-Saharan Cities and the Implications for Urban Agriculture: Evidence-Based Remote Sensing from Niamey, Niger

**Ibrahim Abdoul Nasser and Elhadi Adam ***

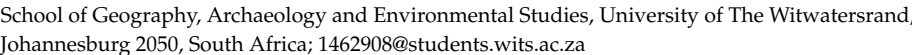

School of Geography, Archaeology and Environmental Studies, University of The Witwatersrand, Johannesburg 2050, South Africa; 1462908@students.wits.ac.za
* Correspondence: elhadi.adam@wits.ac.za; Tel.: +27-117-176-532

**Abstract:** Urbanisation is the process whereby cities are transformed into large sprawling areas. Urbanisation combined with a continuous increase in population makes food security crucial for sustainable development. Urbanisation poses a threat to agricultural land use within built-up and peri-urban areas. It has resulted in the rapid disappearance and/or total change of agricultural farmland in urban and peri-urban areas. To monitor the changes in agricultural farmland, an understanding of changes in the urban landscape is becoming increasingly important. In this study, multi-temporal Landsat imagery were used to analyse the impact of urbanisation on urban agriculture in the city of Niamey. Changes in the urban landscape were determined using the support vector machine (machine learning) algorithm. Results of this study showed a decrease in land with crops from 3428 ha to 648 ha and an increase in built-up areas from 1352 ha to 11,596 ha between 1975 and 2020. Urbanisation and population growth are the main drivers of urban landscape change in Niamey. There was also a decrease in bare land, rock and vegetation classes, while a small increase in rice and water body classes, comparing the 1975 and 2020 values. This study demonstrates the importance of remote sensing in showing the implications of urbanisation on urban agriculture. These results can assist city planners and resource managers in decision-making and adoption of sustainable mitigation measures which are crucial for urban development.

**Keywords:** urbanisation; urban agriculture; urban landscape; Landsat; change detection

## 1. Introduction

Urbanisation is a megatrend that has been experienced by most sub-Saharan countries this century. The demographic definition of urbanisation is the increasing share of a nation's population living in urban areas [1]. This results in a physical expansion of the built environment to house the urban population and their activities [2]. Rapid urbanisation and city growth are caused by several different factors, including rural–urban migration, natural population increase and annexation [3]. However, studies have shown that in most countries around the world, the built environment in urban areas is expanding faster than urban populations [4]. Whereas urban populations were expected to almost double from 2.6 billion in 2000 to 5 billion in 2030 [5,6], urban areas are forecast to triple in extent between 2000 and 2030 [7]. Studies have shown that urban areas offer economies of scale, richer market structures and social development [3,8]. Despite the high rates of urban poverty associated with the cities in developing countries, urban populations enjoy better basic public services such as access to education, health care, electricity, water and sanitation. However, urbanisation in developing countries is causing many challenges such as degradation of the urban environment, habitat damage, social instability and substantial reduction of cultivable lands [9]. The rapid growth of the urban population outstrips most cities' capacity and has increased the demand for urban land, which in turn results in higher land values [9,10]. The increase in urban land value and rent wages are key factors

that contribute to the conversion in less developed and developing countries of urban green spaces and urban agricultural lands into new urban development and peri-urban zones [11,12].

Urban and peri-urban agriculture is defined as the production, processing and distribution of food and other products (plant crops and/or livestock) raised in and around cities to provide urban food security [13]. Urban agricultural lands create buffer zones between cities and natural habitats that provide food, fibre, clean air, soil and water to urban areas. This buffer also reduces the impact of urban systems on the environment and wildlife [14,15]. Studies have estimated that (mainly in developing countries) 200 million urban residents produce food for the urban market, providing 15% to 20% of the world's supply [16]. Despite the importance of this contribution, urban agriculture faces ongoing land insecurity owing to competitive land use [13,15] and unsustainable development of settlements [17]. This is because (particularly in developing countries), city authorities pay little attention to integrating agriculture into the city land use planning and zoning processes. Instead, authorities have emphasised land uses with high bid rents to promote the 'highest and best use' [18]. Owing to their spatial proximity to urban areas, urban agricultural lands are the first ecosystem affected adversely by urban sprawl. Globally, urban agricultural lands have been increasingly transformed into continuous and discontinuous urban growth with the uneven development of infrastructure and settlements [15,19]. In sub-Saharan Africa, it is widely recognised that urbanisation has progressively altered contemporary urban agriculture, negatively influencing ecosystem processes and services, fragmenting or depleting agriculture land systems [15,18–20]. Niger is a sub-Saharan African country covering about 1.27 million square miles of landlocked area. It is one of the world's poorest countries and is recurrently affected by famine and drought periods [21]. The country's capital, Niamey, has been experiencing a high population growth rate of 5.3% per year owing to the high birth rate and large numbers migrating from rural areas into the city [21]. The in-migration to urban areas is contributed to by factors including low soil fertility, erratic rainfall and poor infrastructure that result in low food production and hence poor food security [21]. The city was first settled on the left bank of the Niger River but has progressively expanded on both riverbanks, thus encroaching upon agricultural lands and enticing rural people away from farming [21]. Owing to the increasing demand for growth of the city and expansion of built-up areas, the land in Niamey has become economically attractive, especially along the banks of the Niger River. This has also increased pressure on fertile areas and led to the conversion of arable to building land. The spatial and temporal expansion of Niamey on urban agriculture are still poorly understood [15]. Comprehensive understanding of peri-urban and urban farming and its dynamics over time and space are increasingly required to anticipate possible future trends and contribute to developing effective urban planning policies [20].

The capability of urban farming to feed the growing urban population, especially in developing countries, will depend on detailed spatio-temporal information to enable sustainable urban planning and management [22]. In Niamey, there is still a lack of reliable information on the extent and change of urban and peri-urban land use and land cover (LULC) classes that can be used to promote sustainable urban agriculture [21]. In previous years, costly and time-consuming ground-based mapping techniques were used to capture data on changes in urban LULC classes [23]. However, remote sensing data such as satellite imagery have been a valuable resource in assessing the spatio-temporal dynamics of urban and peri-urban agriculture [24]. Satellite imagery provides cost-effective, up-to-date, accurate and detailed information on LULC classes [25]. The analysis of remote sensing data using robust and reliable change detection algorithms helps analyse the pattern, growth and extent of urbanisation which can enable stakeholders to provide decisions that assist in reducing the negative impacts of urbanisation on the environment [26].

Most previous studies on the impact of urbanisation on urban agriculture were conducted in developed countries [27–30] and a few in Africa [31,32]. However, much of the research conducted in Africa up to now has been descriptive and conducted using costly

survey methods. Previous studies conducted in Niamey have indicated the importance of urban agriculture to the residents [21,33]. According to Bernholt, Kehlenbeck, Gebauer and Buerkert [21], the plant species richness in Niamey changed from 115 species in the cold season, to 110 and 77 species in the hot and rainy seasons, respectively. Graefe, Schlecht and Buerkert [33] highlighted the spatial distribution of urban agriculture activities using ground-based techniques, namely semi-structured interviews. The results from the study indicated that agricultural activities were conducted mainly along the Niger River. However, these previous studies could not cost-effectively analyse the spatio-temporal changes in urban agricultural lands. Past studies in Niamey reported on how urban agriculture has been changing over the years and severely negatively impacted by urbanisation. While it has been demonstrated in other studies that satellite imagery and remote sensing techniques can provide vital information for urban planning and environmental development programmes for achieving sustainable urban agriculture, no studies have been carried out to date to analyse the impact of urbanisation on urban agriculture in Niamey. Hence this study analyses the impact of urbanisation on urban agriculture using remote sensing techniques in Niamey over 45 years.

## 2. Material and Methods

### 2.1. Study Area

Niamey, the capital city of Niger, lies on both banks of the Niger River (Figure 1). The city covers a total area of 239 km², mostly falling on the northern bank of the Niger River [33]. The population of the city is approximately 1,131,882 [34]. The annual average rainfall is around 540 mm per annum, with average temperatures of 33 °C during the hot season and 27 °C in the cold season [35]. Intensive horticulture and millet cropping, as well as milk, meat, rice and egg production are the most common agricultural activities in Niamey [21,33]. The Niger River is the main water source for the irrigation of horticultural crops in the city [21].

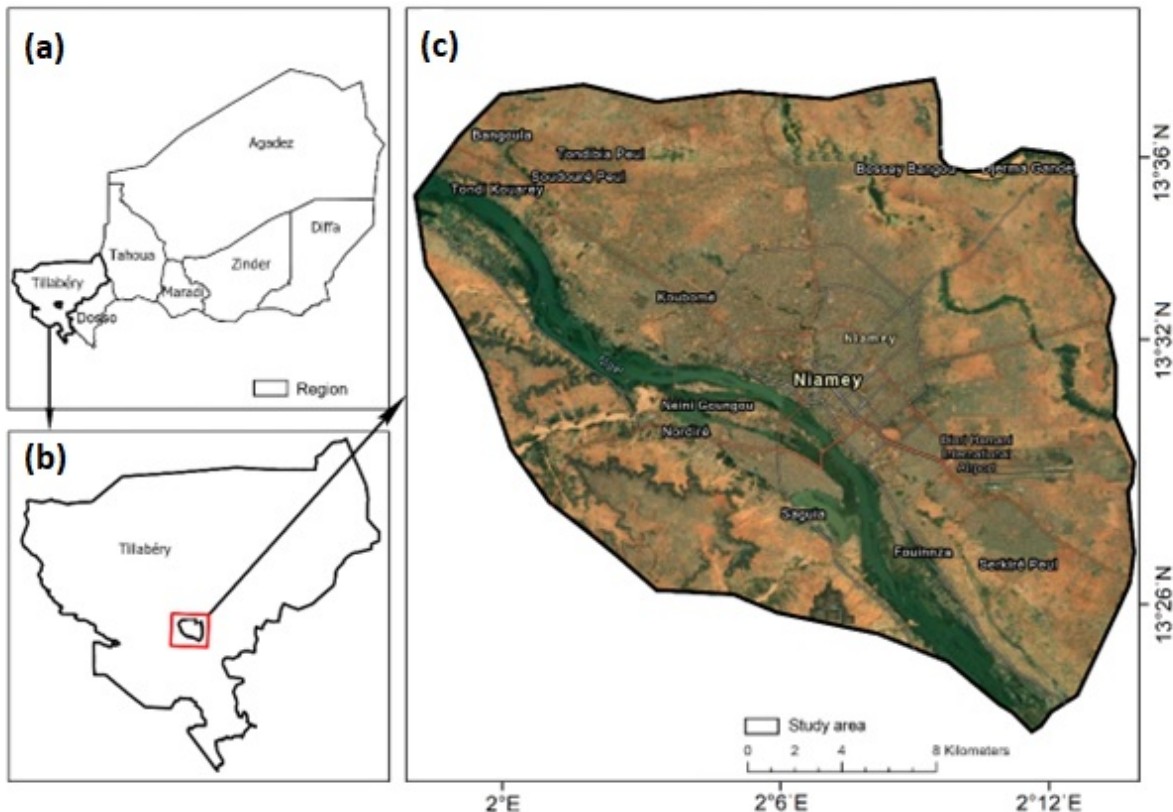

**Figure 1.** (**a**) Administrative regions of Niger, (**b**) Tillabery region and the city of Niamey, and (**c**) Basemap showing the city of Niamey.

## 2.2. Remote Sensing Data Acquisition

Ten cloud-free Landsat images for intervals ranging between three and five years from 1975to 2020 (periods shown in Table 1) were used in this study. The images were obtained from the United States Geological Survey (USGS) data interface (http://www.usgs.gov/, accessed on 26 May 2023). The Landsat datasets were chosen for this study based on the success of previous studies that used them for urban agriculture mapping. They are furthermore freely accessible and provide rich historical data that is suitable for short- and long-term landscape monitoring [36]. The images were acquired during the wet seasons, during which there was high vegetation vigour that enhances the spectral variability between crops and other LULC classes such as built-up, rock and bare land areas. We considered the time interval between the 11 Landsat images adequate to show the changes in urban agriculture.

**Table 1.** Specifications of the Landsat imageries used in this study.

| Satellite | Acquisition Date | Path/Row | Spatial Resolution |
|---|---|---|---|
| Landsat 2 (MSS) | 29 September 1975 | 207/051 | 60 m |
| Landsat 2 (MSS) | 1 November 1979 | 207/051 | 60 m |
| Landsat 5 (TM) | 11 September 1984 | 193/051 | 30 m |
| Landsat 4 (TM) | 30 October 1987 | 193/051 | 30 m |
| Landsat 4 (TM) | 25 September 1992 | 193/051 | 30 m |
| Landsat 5 (TM) | 21 November 1998 | 193/051 | 30 m |
| Landsat 7 (ETM+) | 20 August 2002 | 193/051 | 30 m |
| Landsat 5 (TM) | 10 August 2007 | 193/051 | 30 m |
| Landsat 7 (ETM+) | 18 October 2012 | 193/051 | 30 m |
| Landsat 8 (OLI/TIRS) | 6 September 2017 | 193/051 | 30 m |
| Landsat 8 (OLI/TIRS) | 16 October 2020 | 193/051 | 30 m |

## 2.3. Image Preprocessing

Preprocessing of satellite imagery, whether air- or space-borne, is fundamental to ensuring the accurate spatial location of datasets on the surface of the Earth [37]. The Landsat images were preprocessed to remove the effects that may have arisen from, inter alia, solar zenith angle effects, Earth-Sun distance, topography and temporal changes in target features [38]. Because the images used in this study were captured in different years, their solar radiation differs, which might adversely affect the LULC change detection model. To overcome this issue, the top-of-atmosphere reflectance method which uses the Earth-Sun geometry to adjust for differences in solar irradiance and eliminate solar zenith angle effects was used [39]. Atmospheric correction for all the Landsat images was done using the Fast Line-of-sight Atmospheric Analysis of Hypercubes (FLAASH) model in ENVI 5.4 software. The images for 1975 and 1979 were resampled from a spatial resolution of 60 m to 30 m using the nearest neighbourhood resampling method to match the ones for 1984, 1987, 1992, 1998, 2002, 2007, 2012, 2017 and 2020 imageries. The images were geometrically corrected to a root mean square error (RMSE) of less than 0.5, which is in line with the recommendation by Jensen [40]. A destripe function in ENVI 5.4 software was used to reduce the scan pattern that was caused by scan line shifts on the Landsat 7 (ETM+) imagery for 2012. All the images from 1975 to 2017 were geometrically corrected to the 2020 image using the image-to-image registration tool in ENVI 5.4 software. The image projection used in this study was WGS84/Universal Transverse Mercator (UTM) Zone 31 North.

## 2.4. Urban LULC Classes and Reference Data Collection

The training and reference data for all the maps used in this study were collected from Google historical maps and panchromatic images corresponding to the date the Landsat imageries were acquired. The total data samples per year were split into 70% (training samples) and 30% (test samples) as shown in Table 2. The training samples used to

classify the images and test samples were used to validate the accuracy of the classification model [41,42].

**Table 2.** The number of training (TR) and test samples (TS) used in this study. The urban LULC classes are bare land (BL), built-up (BU), other crops (OC), rice (RC), rock (RK), vegetation (VG) and waterbody (WB).

| Year | 1975 | | 1979 | | 1984 | | 1987 | | 1992 | | 1998 | | 2002 | | 2007 | | 2012 | | 2017 | | 2020 | |
|---|---|---|---|---|---|---|---|---|---|---|---|---|---|---|---|---|---|---|---|---|---|---|
| Class | TR | TS | TR | TS | TR | TS | TR | TS | TR | TS | TR | TS | TR | TS | TR | TS | TR | TS | TR | TS | TR | TS |
| BL | 154 | 66 | 124 | 53 | 154 | 66 | 205 | 88 | 212 | 91 | 261 | 112 | 161 | 69 | 439 | 188 | 182 | 78 | 103 | 44 | 123 | 52 |
| BU | 75 | 32 | 196 | 84 | 63 | 27 | 72 | 31 | 103 | 44 | 91 | 39 | 266 | 114 | 98 | 42 | 152 | 65 | 198 | 85 | 243 | 103 |
| OC | 135 | 58 | 91 | 39 | 107 | 46 | 84 | 36 | 147 | 63 | 107 | 46 | 51 | 22 | 187 | 80 | 168 | 72 | 189 | 81 | 84 | 36 |
| RC | 110 | 47 | 98 | 42 | 152 | 65 | 201 | 86 | 152 | 65 | 156 | 67 | 93 | 40 | 376 | 161 | 75 | 32 | 86 | 37 | 102 | 43 |
| RK | 107 | 46 | 189 | 81 | 86 | 37 | 138 | 59 | 86 | 37 | 84 | 36 | 205 | 88 | 68 | 29 | 159 | 68 | 77 | 33 | 66 | 27 |
| VG | 219 | 94 | 105 | 45 | 79 | 34 | 175 | 75 | 131 | 56 | 126 | 54 | 96 | 41 | 128 | 55 | 138 | 59 | 226 | 97 | 80 | 33 |
| WB | 182 | 78 | 98 | 42 | 168 | 72 | 147 | 63 | 189 | 81 | 264 | 113 | 91 | 39 | 217 | 93 | 133 | 57 | 107 | 46 | 77 | 32 |

*2.5. Image Classification*

A supervised classifier was used to classify and validate the urban LULC classes. The support vector machine (SVM) algorithm, first proposed by Vapnik [43] was used to classify all the images used in this study. The SVM classifier is non-parametric and maximises the margin surrounding the hyperplane that separates the points into different classes [44]. Support vectors are the points that constrain the width of the margin [45]. The hyperplane is found using the formula:

$$y_i(w \times x_i + b) \geq 1 - \xi_i \tag{1}$$

in which $w$ stands for the coefficient vector that determines the orientation of the hyperplane in the feature space [43]. The origin's offset of the hyperplane is represented by $b$; and $\xi_i$ stands for the positive slack variables [43]. The optimal hyperplane is determined by solving the optimisation problem as follows:

$$\text{Minimise} \sum_{i=1}^{n} \alpha_i - \frac{1}{2} \sum_{i-1}^{n} \sum_{j=1}^{n} \alpha_i \alpha_j y_i y_j \left( x_i x_j \right) \tag{2}$$

$$\text{subject to} \sum_{i=1}^{n} \alpha_i y_j = 0, \ 0 \leq \alpha_i \leq C \tag{3}$$

in which $\alpha_i$ stands for the Lagrange multiplier and $C$ represents the penalty [43]. For the classification of linear data, the decision function is applied as follows:

$$g(x) = \text{sign} \left( \sum_{i=1}^{n} y_i \alpha_i x_i + b \right) \tag{4}$$

If the dataset is non-linear, the decision function as shown in Equation (4) is rewritten as follows:

$$g(x) = \text{sign} \left( \sum_{i=1}^{n} y_i \alpha_i K(x_i, x_j) + b \right) \tag{5}$$

In the classification of non-linear data, the dataset is transformed into a higher dimensional space using a kernel function (K). Four K types are commonly used in LULC classification studies, and these are sigmoid, radial basis function, linear and polynomial [46,47]. However, several studies have proven that the radial basis function is superior to other K's in data classification [46–48]. The radial basis function kernel requires the tuning of the 'cost' (C) and the 'gamma' ($\gamma$) parameters, which can affect the classification accuracy [46]. The optimal C and $\gamma$ parameters are selected using a comprehensive search method using a large number of data samples [49]. In this study, the 10-fold cross-validation technique was used to search for the optimal C and $\gamma$ parameters. The optimal C and $\gamma$ parameters used in this study were 120 and 1, respectively.

*2.6. Classification Accuracy*

Confusion matrices were used to assess the classification accuracies of the Landsat imageries used in this study. The matrices compare the true classes with the ones allocated by the SVM classifier on the resultant maps [37]. The classified maps were assessed using Google historical maps and respective panchromatic images. Confusion matrices using the test data samples were then used to compute the kappa statistic, overall, producer's and user's accuracies. The kappa statistic evaluates if there is an agreement between the classifier and the reference data [50]. Overall accuracy is calculated by averaging the correctly classified TS among all the classes [50]. The producer's accuracy determines the possibility of a data sample on the ground is correctly classified, while the user's accuracy ascertains the possibility that a data sample belongs to a particular class and the classifier correctly assigned to it [51].

*2.7. Change Detection*

Change detection analysis in remote sensing studies is a broad process used to identify and quantify differences between two satellite images of the same scene at different times. The univariate image differencing method was applied to all the Landsat imageries in this study. In this method, spatially registered imagesries of time t_2 subtracts t_1, to produce an image that shows changes that happened between the time period. The method is mathematically calculated as follows:

$$Dx_{ij}^k = x_{ij}^k(t_2) - x_{ij}^k(t_1) + C \tag{6}$$

where $t_1$ is the first time, $t_2$ is the second time, $x_{ij}^k$ stands for the pixel value for band *k*. The pixel numbers on the image are represented by *i* and *j*. The constant to produce positive digital numbers in the calculation process is represented by *C*. In this study, the change detection statistics were tabulated showing the changes in images from 1975–1979; 1979–1984; 1984–1987; 1987–1992; 1992–1998; 1998–2002; 2002–2007; 2007–2012; 2012–2017; and 2017–2020. The changes for the LULC classes were expressed in hectares (ha).

## 3. Results

*3.1. Urban Landscape Change*

The classification results showed that most urban farmlands are along the Nile River and built-up structures were mainly situated at the centre of the study area (Figure 2). Limited areas of rice crops, rocks and water bodies were also found in the study area (Figure 2).

There was an increase in built-up area from 1353.34 ha to 20,687 ha and a decrease in area for the other crops category from 2791.55 to 752 ha between 1975 and 2020 (Figure 3). The was an increase in bare land from 18,038 ha to 20,870 ha, while the area covered by vegetation decreased from 5345 ha to 3160 ha from 2017 to 2020 (Figure 3). The urban landscape change in the study area between 1975 and 2020 is shown in Figure 3.

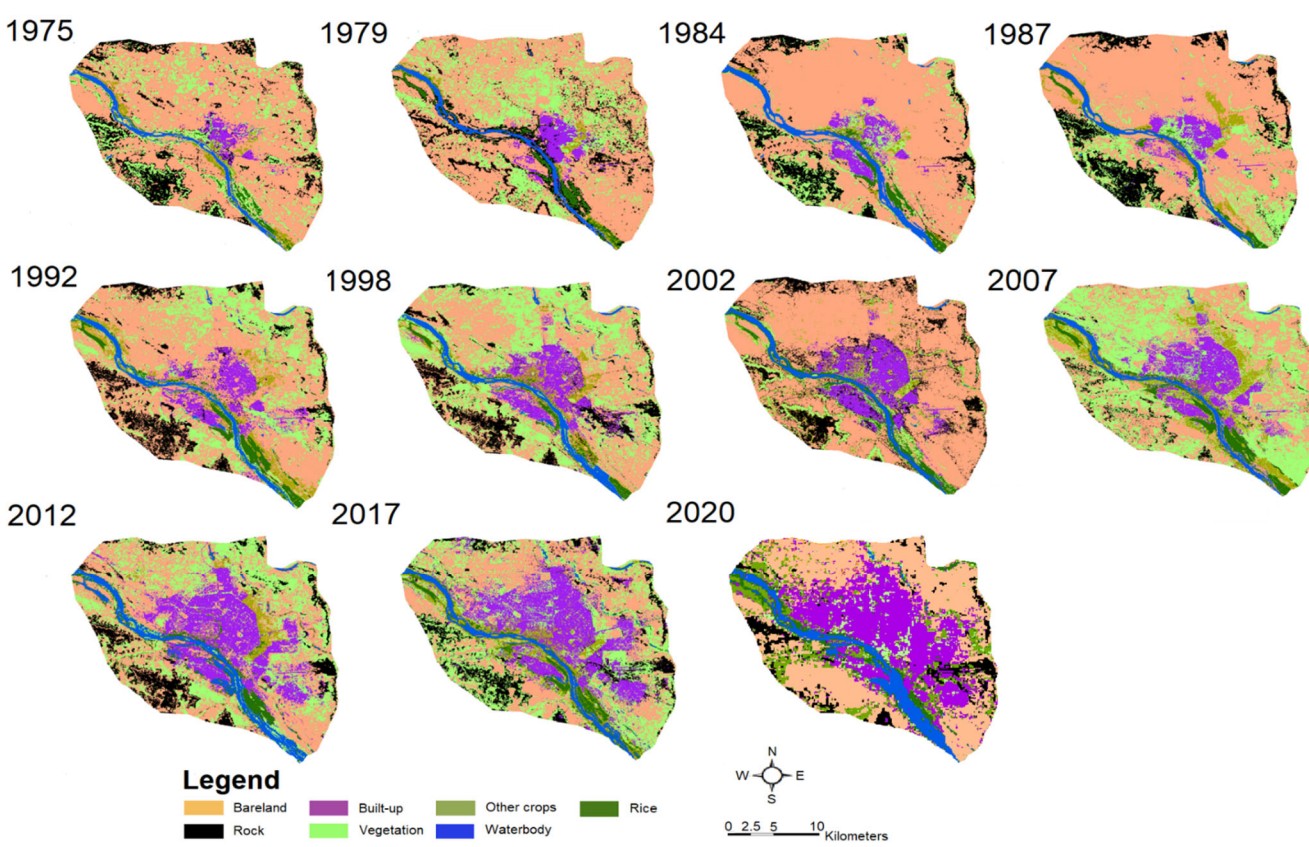

**Figure 2.** Urban LULC maps for Niamey from 1975 to 2020.

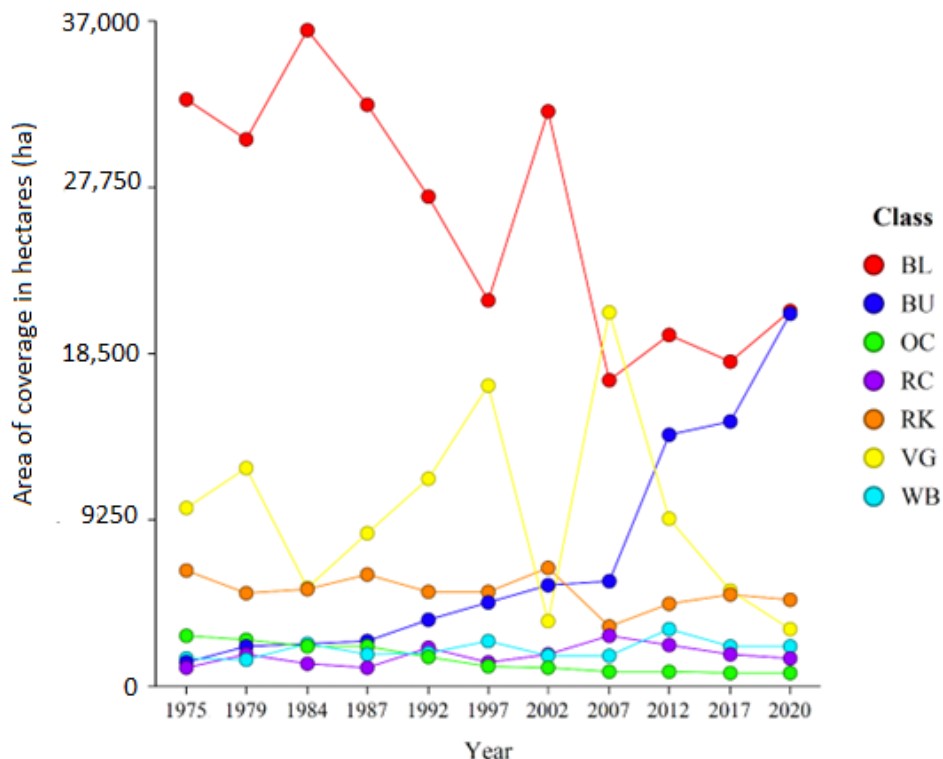

**Figure 3.** Urban LULC distribution for the city of Niamey between 1975 to 2020. The LULC classes are bare land (BL), built-up (BU), other crops (OC), rice (RC), rock (RK), vegetation (VG) and waterbody (WB).

### 3.2. Inter-Annual Urban Landscape Change

The highest urban landscape change was from vegetation to bare land, with a value of 15 888 ha between 2002 and 2007, while the lowest changes were from water bodies to built-up, other crops and the rock class in the year ranges 1975–1979, 1984–1987, 2007–2012, 2012–2017 and 2017–2020 (Table 3). The detailed inter-annual and interclass urban landscape variability over the years of study are provided in Table 3. The urban landscape per class in the years of study is shown in Figure 4. The highest urban landscape change (−402%) was for the rock class between 2002 and 2007 (Figure 4). There were positive changes for the built-up class in all the years of study with the highest value of 88% in 2020 (Figure 4). There was an increase in built-up areas and a reduction in urban agriculture (Figure 5). The highest value (11,596 ha) was for the built-up class and the lowest value (648 ha) was for urban agriculture in 2020 (Figure 5). The changes in urban agriculture and built-up over the years in Niamey are shown in Figure 5.

**Table 3.** Change detection statistics of urban LULC classes for Niamey between 1975 and 2020. The LULC classes are bare land (BL), built-up (BU), other crops (OC), rice (RC), rock (RK), vegetation (VG) and waterbody (WB).

| CLASS | BL | BU | OC | RC | RK | VG | WB |
|---|---|---|---|---|---|---|---|
| | | | | 1975–1979 | | | |
| BL | 19,332 | 1002 | 235 | 185 | 1549 | 10,206 | 144 |
| BU | 307 | 886 | 18 | 2 | 58 | 81 | 0 |
| OC | 212 | 64 | 287 | 265 | 250 | 30 | 3 |
| RC | 33 | 1 | 76 | 670 | 232 | 0 | 8 |
| RK | 5481 | 117 | 9 | 16 | 440 | 400 | 2 |
| VG | 4966 | 122 | 139 | 644 | 2600 | 1415 | 40 |
| WB | 85 | 21 | 21 | 18 | 63 | 0 | 1311 |
| | | | | 1979–1984 | | | |
| BL | 21,578 | 1074 | 219 | 43 | 4266 | 3055 | 180 |
| BU | 657 | 1010 | 39 | 11 | 17 | 433 | 46 |
| OC | 308 | 5 | 156 | 59 | 14 | 119 | 126 |
| RC | 536 | 13 | 59 | 817 | 17 | 119 | 242 |
| RK | 2434 | 105 | 233 | 333 | 905 | 880 | 303 |
| VG | 10,915 | 172 | 50 | 1 | 153 | 835 | 1 |
| WB | 23 | 2 | 1 | 4 | 0 | 7 | 1471 |
| | | | | 1984–1987 | | | |
| BL | 28,022 | 543 | 944 | 236 | 1480 | 5202 | 25 |
| BU | 814 | 1327 | 34 | 8 | 3 | 196 | 1 |
| OC | 125 | 56 | 129 | 13 | 2 | 427 | 0 |
| RC | 347 | 66 | 229 | 395 | 11 | 210 | 4 |
| RK | 1346 | 30 | 20 | 7 | 3434 | 539 | 0 |
| VG | 1547 | 454 | 233 | 49 | 1295 | 1869 | 4 |
| WB | 151 | 21 | 30 | 320 | 2 | 77 | 1769 |
| | | | | 1987–1992 | | | |
| BL | 17,316 | 1 895 | 1124 | 569 | 1123 | 10,183 | 141 |
| BU | 636 | 1512 | 70 | 83 | 97 | 66 | 32 |
| OC | 766 | 14 | 440 | 355 | 6 | 37 | 2 |
| RC | 116 | 8 | 25 | 680 | 1 | 142 | 55 |
| RK | 2219 | 38 | 12 | 14 | 3578 | 355 | 10 |
| VG | 6200 | 262 | 582 | 386 | 448 | 581 | 63 |
| WB | 17 | 1 | 1 | 48 | 3 | 200 | 1534 |

**Table 3.** *Cont.*

| CLASS | BL | BU | OC | RC | RK | VG | WB |
|---|---|---|---|---|---|---|---|
| | | | | 1992–1998 | | | |
| BL | 15,993 | 1510 | 1072 | 185 | 966 | 7432 | 111 |
| BU | 492 | 2520 | 130 | 10 | 148 | 397 | 34 |
| OC | 1118 | 122 | 505 | 117 | 20 | 347 | 26 |
| RC | 238 | 44 | 410 | 922 | 65 | 231 | 222 |
| RK | 730 | 158 | 8 | 8 | 3274 | 1079 | 1 |
| VG | 2875 | 279 | 83 | 44 | 765 | 7141 | 376 |
| WB | 9 | 6 | 6 | 14 | 3 | 70 | 1730 |
| | | | | 1998–2002 | | | |
| BL | 16,223 | 906 | 1035 | 71 | 2075 | 1135 | 9 |
| BU | 393 | 3414 | 233 | 31 | 415 | 152 | 2 |
| OC | 353 | 383 | 589 | 200 | 95 | 589 | 5 |
| RC | 56 | 67 | 164 | 598 | 10 | 397 | 7 |
| RK | 1274 | 287 | 480 | 14 | 2192 | 995 | 2 |
| VG | 13,538 | 480 | 247 | 287 | 1776 | 299 | 68 |
| WB | 99 | 78 | 43 | 612 | 16 | 75 | 1577 |
| | | | | 2002–2007 | | | |
| BL | 12,387 | 967 | 2168 | 250 | 203 | 15,888 | 38 |
| BU | 393 | 3814 | 462 | 179 | 6 | 773 | 5 |
| OC | 323 | 258 | 544 | 304 | 183 | 1194 | 3 |
| RC | 350 | 12 | 56 | 1210 | 0 | 45 | 152 |
| RK | 3178 | 646 | 359 | 24 | 283 | 2083 | 2 |
| VG | 232 | 126 | 398 | 835 | 637 | 1398 | 11 |
| WB | 127 | 0 | 2 | 38 | 0 | 10 | 1490 |
| | | | | 2007–2012 | | | |
| BL | 3426 | 1592 | 76 | 118 | 2650 | 8711 | 412 |
| BU | 493 | 4916 | 15 | 61 | 21 | 276 | 19 |
| OC | 1963 | 544 | 609 | 386 | 13 | 380 | 74 |
| RC | 183 | 160 | 3 | 1212 | 10 | 274 | 987 |
| RK | 203 | 18 | 0 | 5 | 1147 | 33 | 0 |
| VG | 13,294 | 2081 | 322 | 527 | 880 | 4203 | 67 |
| WB | 2 | 8 | 0 | 10 | 0 | 37 | 1625 |
| | | | | 2012–2017 | | | |
| BL | 6290 | 87 | 249 | 61 | 460 | 9921 | 14 |
| BU | 22 | 7593 | 952 | 873 | 8 | 312 | 70 |
| OC | 8843 | 1 475 | 338 | 103 | 888 | 2198 | 35 |
| RC | 11 | 33 | 361 | 692 | 5 | 79 | 55 |
| RK | 231 | 90 | 24 | 2 | 3344 | 940 | 0 |
| VG | 255 | 2167 | 377 | 56 | 412 | | 23 |
| WB | 216 | 142 | 128 | 1 | 18 | 523 | 2026 |
| | | | | 2017–2020 | | | |
| BL | 4290 | 2 984 | 1698 | 45 | 352 | 5823 | 52 |
| BU | 5765 | 10,525 | 425 | 1063 | 106 | 215 | 12 |
| OC | 1362 | 265 | 555 | 61 | 25 | 563 | 56 |
| RC | 11 | 44 | 153 | 691 | 5 | 85 | 5 |
| RK | 587 | 152 | 12 | 3 | 3758 | 26 | 0 |
| VG | 214 | 5469 | 1087 | 55 | 326 | 2603 | 55 |
| WB | 116 | 124 | 128 | 1 | 18 | 52 | 2025 |

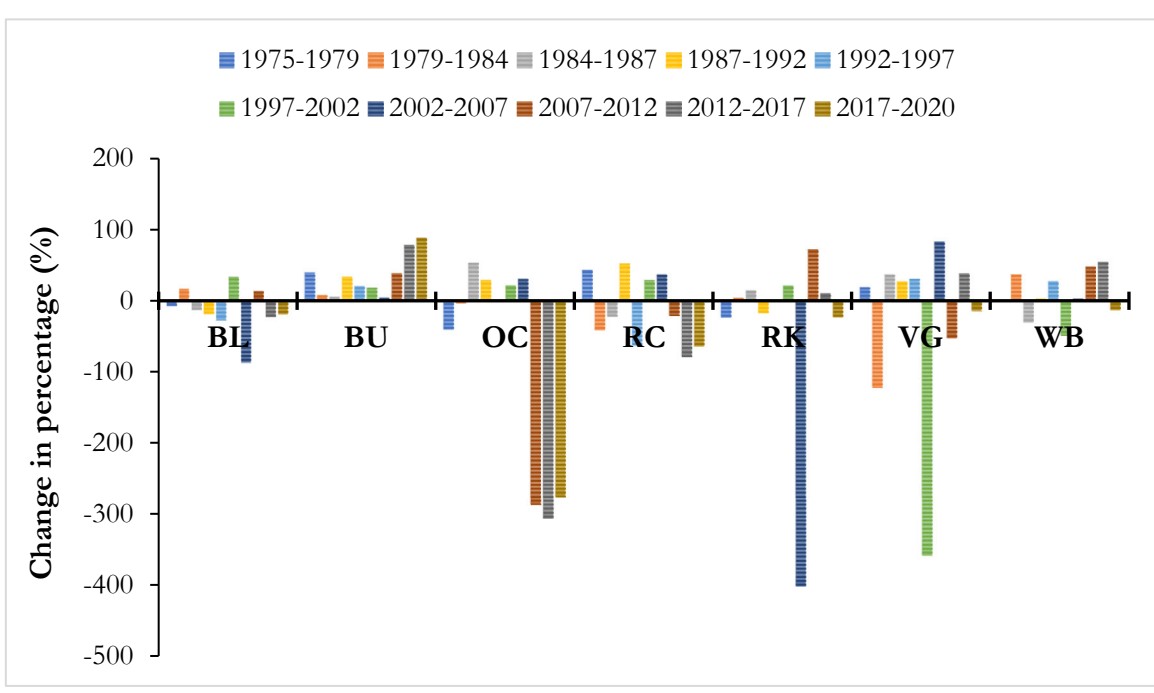

**Figure 4.** Changes in LULC classes between 1975 and 2020 in Niamey. The LULC classes are (BL = Bare land, BU = Built-up, OC = Other crops, RC = Rice, RK = Rock, VG = Vegetation, and WB = Waterbody).

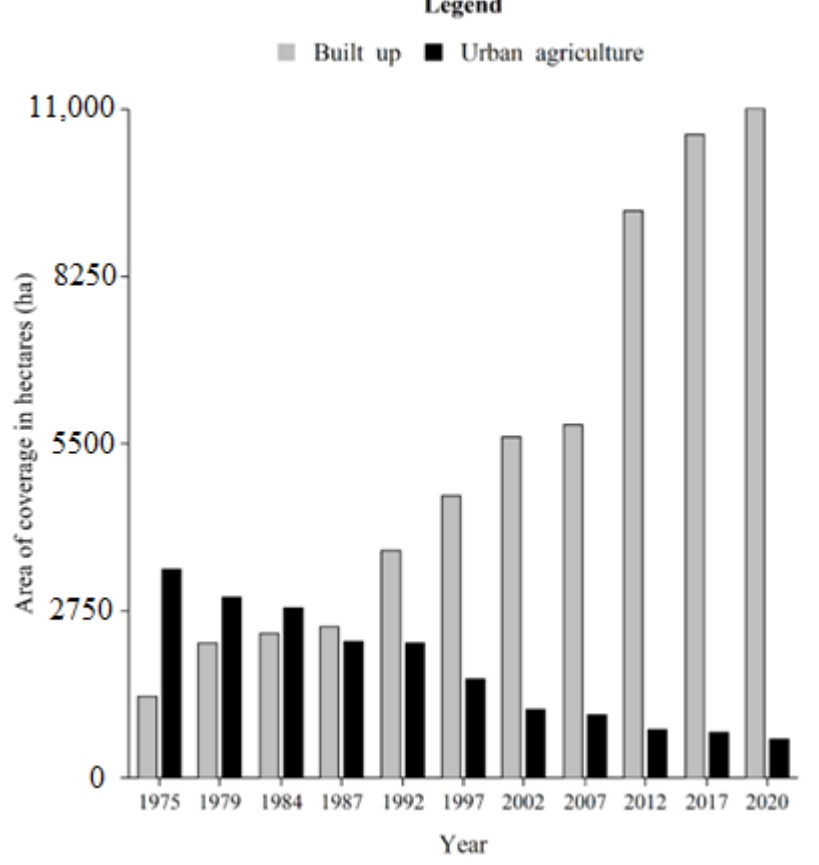

**Figure 5.** Changes in the extent of built-up and urban agriculture from 1975 to 2020 in hectares (ha).

### 3.3. Validation

The overall accuracies for all the maps were greater than 85%, which is a threshold value recommended by Anderson [52] in LULC classification, as shown in Table 3. The kappa coefficients for all the classified maps were all above 80%, with the highest value of 96% for 2020, and the lowest value of 84% for 1975 (Table 4). The user's accuracies had the lowest value of 59% for the built-up class in 1975 (Table 4). The user's accuracies for the water body class were all ≥ 95% (Table 4). The producer's accuracies produced the lowest value of 59% for the built-up class in 1975, while the values for the waterbody class were all ≥ 99% (Table 3).

**Table 4.** Confusion matrices showing overall accuracy (OA), producer's accuracy (PA) and Kappa coefficient value for 1975, 1979, 1984, 1987, 1992, 1998, 2002, 2007, 2012, 2017 and 2020. The LULC classes are (BL = Bare land, BU = Built-up, OC = Other crops, RC = Rice, RK = Rock, VG = Vegetation, and WB = Waterbody).

| | Class | | BL | | BU | | OC | | RC | | RK | | VG | | WB | |
|---|---|---|---|---|---|---|---|---|---|---|---|---|---|---|---|---|
| | Type | | UA | PA | UA | PA | UA | PA | UA | PA | UA | PA | UA | PA | UA | PA |
| Year | OA | Kappa | | | | | | | | | | | | | | |
| 1975 | 87% | 84 | 89 | 83 | 59 | 59 | 78 | 85 | 98 | 98 | 91 | 83 | 85 | 85 | 96 | 100 |
| 1979 | 88% | 85 | 86 | 93 | 94 | 88 | 90 | 69 | 84 | 91 | 83 | 86 | 78 | 84 | 100 | 100 |
| 1984 | 93% | 91 | 97 | 96 | 83 | 93 | 86 | 80 | 100 | 100 | 97 | 95 | 70 | 77 | 100 | 99 |
| 1987 | 95% | 94 | 99 | 100 | 91 | 94 | 83 | 83 | 99 | 98 | 95 | 88 | 87 | 91 | 100 | 100 |
| 1992 | 93% | 91 | 90 | 98 | 95 | 84 | 91 | 78 | 100 | 100 | 94 | 92 | 80 | 90 | 98 | 100 |
| 1998 | 94% | 92 | 97 | 96 | 83 | 90 | 79 | 89 | 98 | 93 | 95 | 97 | 92 | 82 | 98 | 99 |
| 2002 | 92% | 90 | 100 | 100 | 96 | 99 | 48 | 46 | 93 | 95 | 99 | 94 | 68 | 68 | 100 | 100 |
| 2007 | 91% | 89 | 99 | 88 | 80 | 98 | 92 | 83 | 98 | 99 | 67 | 90 | 67 | 73 | 95 | 99 |
| 2012 | 91% | 89 | 95 | 95 | 90 | 95 | 91 | 79 | 83 | 91 | 94 | 93 | 79 | 83 | 100 | 100 |
| 2017 | 93% | 92 | 98 | 98 | 99 | 94 | 85 | 89 | 94 | 84 | 92 | 100 | 90 | 91 | 100 | 100 |
| 2020 | 96% | 96 | 96 | 98 | 98 | 96 | 94 | 94 | 93 | 98 | 96 | 96 | 94 | 91 | 100 | 100 |

### 4. Discussion

The multi-temporal urban landscape change analysis on the eleven Landsat imageries shows that there have been LULC changes in Niamey (Figure 3). There was an increase in the built-up areas and a decrease in areas with rice, as well as with other crops (Figure 3). This is due to population increase where agricultural land is cleared for built-up structures. This is in line with previous studies [53–55] which note that there has been an increase in settlements owing to population growth. This has resulted in changes in other LULC classes such as urban agriculture land. The population growth in Niamey is supported by Statista [56] which highlights that the city is one of the fastest-growing cities in Africa with an estimated growth rate of 101%, which has an impact on its economy and agricultural activities. The decrease in the land covered by rice, as well as the category of other crops, is in line with Balineau, et al. [57] who noted that there has been a reduction in urban agriculture in Niamey owing to competition between agricultural land and other productive sectors and residential uses.

The major noticeable change was from vegetation to bare land—with a value of 15 888 ha between 2002 and 2007 (Table 3 and Figure 4). The loss of vegetation cover in the study area between 2003 and 2007 was caused by low rainfall and urbanisation (where trees were cut to clear the land for built-up structures) [58]. The loss of land by 23%, 306% and 79% for bare land, other crops, and vegetation, respectively (Figure 4), is attributed to urbanisation in the city. This is supported by Salamatou, Abdoulaye, Boubacar, Abou Soufianou, Ali and Mahamane [54] who stated that the growing spatial demand from the rising urban population in Niamey has led to a decrease in the area covered by crops, bare soil and vegetation. The rise in population has also been due to the in-migration of people from rural areas in Niger because of repeated droughts and food crises [33]. There are high fertility rates and declining mortality rates in Niamey which has created pressure on the

city because of the increase in built-up structures [59]. The construction of buildings on the outskirts of the city has also resulted in the loss of land for crop cultivation [59].

There was massive vegetation loss in Niamey between 1997 and 2002 (Figure 4). The loss of this vegetation was due to agricultural land clearing, grazing, climate change and urban fuel demand [60]. This was the result of a high rate of in-migration stemming from environmentally induced economic movement leading to deforestation to clear land for agriculture and settlements [61]. The clearing of land was mainly done along the Niger River where agricultural fields are mainly located and this has resulted in exacerbation of flooding events owing to increased runoff [62]. There was a loss of vegetation (−123%) between 1997 and 2002, as shown in Figure 4, which was a result of repeated droughts [55]. However, there were increases in the extent of vegetation during the years 1975–1979, 1984–1987, 2002–2007 and 2012–2017. The increase in vegetation is due to tree planting and management techniques. Some of the tree planting programmes such as Operation Sahel Vert (Operation Green Sahel) and Fête nationale de l'Arbre (National Tree Day) launched by Seyni Kountché in 1975 encouraged youth people to plant trees across Niger to reduce desertification and extend his rule across all the country's territories and populations [63]. This led to an increase in vegetation across the city, and over 60 million trees were planted across the country [55].

There was an inverse change in the area covered by built-up structures and crops (Figure 5). The decrease in the area with crops has been one of the main issues in Niamey and it has resulted in food insecurity in the city. The National Adaptation Programme Action (NAPA) and Plan National de l'Environnement pour un Développement Durable (PNEDD) formed to reduce desertification and improve the daily lives of people in Niger identified that the decrease in land with crops is due to climate hazards such as droughts, extreme temperatures, strong winds, flooding dust storms and insect infestations [59]. These climate hazards have hurt the economy with a decrease in fishery productivity, groundwater depletion, formation of dunes and increased death of livestock. The increase in the population in Niamey is due to the in-migration of many young men into the city to work for their families; they move to the city in the hope of finding employment, and to ward off poverty and food insecurity [64]. The results of this study correspond to the United Nations [65] study which noted that the population in Niamey grew from 198,099 in 1975 to approximately 1.3 million residents in 2020.

The OA results for all the classified maps in this study were high, with all of them greater than 85% (Table 3). The high accuracies were due to the SVM classifier's strength in handling numerical data without assuming data distribution, and its ability to derive sets of valuable features from all the classes to characterise the urban environment [26,66]. The user's and producer's accuracies for the maps were high with maximum values of 100% (Table 3). This was due to the classifier's ability to deal with noisy data where outliers effects were encountered because of atmospheric and topographic distortions. On the other hand, the lowest user's and producer's accuracies (59%) were for the built-up class in 1975. This was because the 1975 image was resampled from 60 m to 30 m pixel size using the nearest neighbourhood resampling method, which reduces the image quality and the image's geolocation accuracy [67].

The results of this study show the importance of remote sensing in showing the effects of urbanisation on urban agriculture over the years. This can assist city planners, farmers, economists, policymakers and resource managers in making effective decisions for sustainable urban development in the rapidly changing city. This study was done at the city level, and it would be of greater value if scaled up to the district or national level to analyse the effects of urbanisation on the cities which can be used to create an urban farming inventory. The urban farming inventory can be used to reduce food insecurities and poverty in the country. The SVM classifier and the free Landsat images used in this study can provide planners, farmers and the government with a powerful toolset to analyse assess and evaluate the spatio-temporal changes of the urban landscape at the local level

or elsewhere. They can also assist roleplayers to understand the drivers responsible for the changes.

### 5. Conclusions

This study contributes to the growing scientific literature on the use of low-cost remote sensing methods in analysing the impacts of urbanisation on urban agriculture over a long period. In this study, a detailed analysis of the impact of urbanisation in Niamey (Niger) over 45 years was performed and the results were presented in urban LULC maps. This study shows that urbanisation has caused significant changes in the landscape from 1975 to 2020. Specifically, there was a decline in bare land, rice, other crops, rocks, vegetation and water bodies and an increase in the built-up areas. The use of remote sensing methods and ancillary datasets is important in understanding the spatio-temporal urban LULC changes. Research into such changes at the city level is of paramount importance as it forms a critical basis for decision-making, policy formulation and administration for sustainable urban development.

**Author Contributions:** Conceptualization, I.A.N. and E.A.; methodology, I.A.N. and E.A.; software, I.A.N.; validation, I.A.N., and E.A.; formal analysis, I.A.N.; data curation, I.A.N.; writing—original draft preparation, I.A.N.; writing—review and editing, I.A.N. and E.A.; visualization, E.A.; supervision, E.A. All authors have read and agreed to the published version of the manuscript.

**Funding:** This research received no external funding.

**Data Availability Statement:** The data will be available upon request.

**Acknowledgments:** We would like to express our sincere gratitude to the University of the Witwatersrand for providing the necessary infrastructure and resources for conducting this research. We are also deeply indebted to the open-source R software community for their invaluable contribution to data analysis. The accessibility and versatility of R software significantly enhanced our ability to process and interpret our research findings. Furthermore, we extend our heartfelt thanks to Simbarashe Jombo for his exceptional assistance with writing. Jombo's expertise and guidance were instrumental in shaping and refining the manuscript. His dedication to improving the quality of our work is deeply appreciated.

**Conflicts of Interest:** The authors declare no conflict of interest.

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
