# Peer review of "Urbanisation in Sub-Saharan Cities and the Implications for Urban Agriculture: Evidence-Based Remote Sensing from Niamey, Niger"

_urbansci, doi:10.3390/urbansci8010005_

Round 1
Reviewer 1 Report
Comments and Suggestions for Authors
The methodology is comprehensive and follows a systematic approach to urban land use and land cover classification.
The use of Landsat imageries from various years provides a temporal perspective, allowing for the analysis of changes over time.
Geometric correction and the use of ENVI 5.4 software ensure the accuracy and reliability of the data.
The division of data into training and test samples is a standard practice in remote sensing and ensures the robustness of the classification model.
The detailed breakdown of author contributions provides transparency regarding the roles and responsibilities of each contributor.
Suggestions:
1. Multi-source Data Integration:
Rationale: Using data from a single source, such as Landsat, might limit the depth and breadth of the analysis.
Improvement: Incorporate data from other satellite sources like Sentinel, ASTER, or MODIS to provide a more comprehensive view and to cross-validate findings.
2. Advanced Pre-processing Techniques:
Rationale: Basic geometric corrections and destriping are essential, but there might be other distortions or noise in the data.
Improvement: Implement atmospheric correction techniques to mitigate the effects of atmospheric scattering and absorption. Also, consider using radiometric calibration to ensure consistent reflectance values across different dates.
3. Machine Learning and Deep Learning Approaches:
Rationale: Traditional classification methods might not capture the complexity of urban landscapes.
Improvement: Utilize machine learning algorithms like Random Forest or Support Vector Machines for classification. For even more advanced analysis, consider deep learning techniques like Convolutional Neural Networks (CNNs) which have shown promise in remote sensing applications.
4. Temporal Analysis:
Rationale: The study spans several years, but it's unclear if temporal dynamics are analyzed.
Improvement: Implement change detection techniques to understand how urban land use and land cover have evolved over time. This can provide insights into urbanization trends, deforestation rates, and more.
5. Ground Truthing:
Rationale: Relying solely on Google historical maps and panchromatic images might introduce errors.
Improvement: Conduct field surveys to collect ground truth data. This can significantly improve the accuracy of the classification and provide validation for the remote sensing data.
6. Incorporate Ancillary Data:
Rationale: Remote sensing data alone might not capture all nuances of urban development.
Improvement: Integrate ancillary data such as topographic maps, soil maps, and socio-economic data to provide a more holistic understanding of urban land use patterns.
7. Uncertainty Analysis:
Rationale: Every classification method has inherent uncertainties.
Improvement: Conduct an uncertainty analysis to understand the potential errors in the classification. This can be achieved through techniques like error matrices, kappa statistics, and bootstrapping.
8. Stakeholder Engagement:
Rationale: Understanding urban land use and land cover has implications for urban planning, policy-making, and environmental conservation.
Improvement: Engage with local stakeholders, urban planners, and policymakers to ensure the research addresses real-world challenges and can be applied in practical scenarios.
9. Expand the Study Area:
Rationale: A larger study area can provide more comprehensive insights.
Improvement: Consider expanding the study to other urban regions or comparing multiple cities to understand broader trends and patterns.
10. Feedback Loop:
Rationale: Continuous improvement is key in research.
Improvement: After the initial study, revisit the methodology based on feedback from peers, stakeholders, and the results themselves. Refine the approach and reapply it to ensure better outcomes in subsequent studies.
Incorporating these suggestions can enhance the robustness and applicability of the study, making it more valuable for both academic and practical purposes
Comments on the Quality of English Language
Overall, comprehensive
Author Response
Urbanisation in sub-Saharan cities and the implications for urban agriculture; evidence-based remote sensing from Niamey, Niger
Reviewer 1
General Comments for Authors
The methodology is comprehensive and follows a systematic approach to urban land use and land cover classification.
The use of Landsat imageries from various years provides a temporal perspective, allowing for the analysis of changes over time.
Geometric correction and the use of ENVI 5.4 software ensure the accuracy and reliability of the data.
The division of data into training and test samples is a standard practice in remote sensing and ensures the robustness of the classification model.
The detailed breakdown of author contributions provides transparency regarding the roles and responsibilities of each contributor.
Responses: We the reviewers for the positive comments.
Suggestions:
- Multi-source Data Integration:
Rationale: Using data from a single source, such as Landsat, might limit the depth and breadth of the analysis. Improvement: Incorporate data from other satellite sources like Sentinel, ASTER, or MODIS to provide a more comprehensive view and to cross-validate findings.
Response
We agree with the reviewer's assertion that incorporating multisource remotely sensed data enhances the robustness and reliability of the findings. Nevertheless, it is important to note that our study's primary objective is to comprehensively analyse urban expansion from 1975 to 2020, and during this period, only the Landsat Sensor has full archival data coverage.
- Advanced Pre-processing Techniques:
Comments
Rationale: Basic geometric corrections and de striping are essential, but there might be other distortions or noise in the data.Improvement: Implement atmospheric correction techniques to mitigate the effects of atmospheric scattering and absorption. Also, consider using radiometric calibration to ensure consistent reflectance values across different dates.
Response:
atmospheric correction and Landsat 7 Destriping have been further explained on page 5 section 2.3
- Machine Learning and Deep Learning Approaches:
Rationale: Traditional classification methods might not capture the complexity of urban landscapes.
Improvement: Utilize machine learning algorithms like Random Forest or Support Vector Machines for classification. For even more advanced analysis, consider deep learning techniques like Convolutional Neural Networks (CNNs) which have shown promise in remote sensing applications.
Response
We appreciate your valuable input and the suggestion to explore deep learning Convolutional Neural Networks (CNNs) for our study focused on urban expansion mapping. Indeed, CNNs have demonstrated promise in remote sensing applications and have the potential to enhance the accuracy and efficiency of our results.
However, we'd like to provide some context for our choice of the Support Vector Machine (SVM) in this study. Our primary objective is to map urban expansion from 1975 to 2020, which is a substantial temporal range. The SVM is a well-established machine learning algorithm that has proven to be effective in analysing such data, particularly when dealing with multispectral and multitemporal imagery.
While CNNs are undoubtedly a promising approach, they often require large volumes of data for training and can be more suitable for more detailed classification tasks, such as object recognition.
That said, we appreciate your suggestion and acknowledge that future work could indeed explore the integration of CNNs, especially if more diverse and extensive datasets become available. We will consider this approach for potential follow-up studies with different data sources or specific objectives.
- Temporal Analysis:
Rationale: The study spans several years, but it's unclear if temporal dynamics are analyzed.
Improvement: Implement change detection techniques to understand how urban land use and land cover have evolved over time. This can provide insights into urbanization trends, deforestation rates, and more.
Response
Change detection has further been explained in the method (2.7) and results (3.2) sections
- Ground Truthing:
Rationale: Relying solely on Google historical maps and panchromatic images might introduce errors.
Improvement: Conduct field surveys to collect ground truth data. This can significantly improve the accuracy of the classification and provide validation for the remote sensing data.
Response:
We appreciate your suggestion and understand the importance of ground truth data in remote sensing and image classification processes. However, our rationale for relying on Google historical maps and panchromatic images as our primary data sources. The challenge we faced in this study was the availability of ground truth data for the historical image from 1975. While ground surveys are undoubtedly a reliable method for validation, obtaining historical ground data for that specific time frame has proven to be exceptionally difficult and, in many cases, infeasible.
Our decision to utilize Google historical maps and panchromatic images was based on their accessibility and coverage. These resources have been widely used in remote sensing and land cover classification studies, and they have proven to be effective for broad-scale analysis.
Additionally, as you pointed out, many studies have successfully employed similar methods using historical map data.
We genuinely value your feedback, and while we acknowledge the potential benefits of field surveys and ground data collection, we believe that our approach is appropriate given the limitations in obtaining historical ground truth data. We have taken care to validate our methodology with the available resources to ensure the robustness of our results.
- Incorporate Ancillary Data:
Rationale: Remote sensing data alone might not capture all nuances of urban development.
Improvement: Integrate ancillary data such as topographic maps, soil maps, and socio-economic data to provide a more holistic understanding of urban land use patterns.
Response
We agree with the reviewer. However, Niamey lacks reliable information and spatial data infrastructure.
- Uncertainty Analysis:
Rationale: Every classification method has inherent uncertainties.
Improvement: Conduct an uncertainty analysis to understand the potential errors in the classification. This can be achieved through techniques like error matrices, kappa statistics, and bootstrapping.
Response
We agree with the reviewer that uncertainty analysis is essential for image classification before proceeding to the change detection. Accuracy assessment has been explained in section 2.6 and confusion matrices, kappa statistics and other analyses have been added.
- Stakeholder Engagement:
Rationale: Understanding urban land use and land cover has implications for urban planning, policy-making, and environmental conservation.
Improvement: Engage with local stakeholders, urban planners, and policymakers to ensure the research addresses real-world challenges and can be applied in practical scenarios.
Response
We thank the reviewer for this valuable suggestion. This manuscript is part of a research project looking at the impact of urbanization on urban agriculture and food security. Engaging with local stakeholders, urban planners, and policymakers on food security is an essential aspect of the research project. We are currently busy collecting data from the community and local stakeholders for the second manuscript.
- Expand the Study Area:
Rationale: A larger study area can provide more comprehensive insights.
Improvement: Consider expanding the study to other urban regions or comparing multiple cities to understand broader trends and patterns.
Response
Niamey, the capital city of Niger, has experienced significant urbanization in recent years. The city's population growth and expansion have led to changes in land use patterns and increased pressure on urban agriculture. This rapid urbanization presents an ideal case study for examining the effects of urban growth on agricultural practices within urban areas. Studying this dynamic in the city allows us to assess how changes in land use and urban expansion affect the socio-economic well-being of urban farmers and their communities. We believe that findings from Niamey can provide valuable insights applicable to a broader context.
- Feedback Loop:
Rationale: Continuous improvement is key in research.
Improvement: After the initial study, revisit the methodology based on feedback from peers, stakeholders, and the results themselves. Refine the approach and reapply it to ensure better outcomes in subsequent studies. Incorporating these suggestions can enhance the robustness and applicability of the study, making it more valuable for both academic and practical purposes.
Response
We appreciate your interest and feedback, and we remain committed to conducting a thorough and informative study in Niamey.

Reviewer 2 Report
Comments and Suggestions for Authors
The article analyzes changes in land use in the city of Niamey. They are due to urbanization to the detriment of urban agriculture. But the methodology is a bit complex and the Figures 3 and 4 on changes in LUCC distribution do not provide clear explanations of the processes at work.
The question arises as to how such remote sensing studies can help stakeholders make decisions for sustainable urban development.
It would be necessary to take again the article to be more convincing.
There is a little mistake in the paragraph 1.2, line132: the landscape images are from 1975 (and non 1979) to 2020.
Author Response
Reviewer 2
The article analyzes changes in land use in the city of Niamey. They are due to urbanization to the detriment of urban agriculture. But the methodology is a bit complex and the
Figures 3 and 4 on changes in LUCC distribution do not provide clear explanations of the processes at work.
Response:
The description of Fig 3 and 4 has been improved.
The question arises as to how such remote sensing studies can help stakeholders make decisions for sustainable urban development.
Response:
Remote sensing studies offer a wealth of information and tools that can greatly benefit stakeholders and decision-makers involved in urban development. Remote sensing provides access to a rich source of spatial data, including land cover, land use, and environmental variables. By analysing this data, stakeholders can make informed decisions related to urban planning, infrastructure development, and resource allocation. Our research, which focuses on urban agriculture and its interaction with urban expansion, can provide valuable insights into the dynamics of land use changes over time. Stakeholders can use this information to optimize urban expansion plans, safeguard green spaces, and promote balanced land use practices. Remote sensing can contribute to environmental impact assessment, urban resilience and disaster management, infrastructure planning, equity and social inclusion etc.
In the discussion section, we have added the role of remote sensing in monitoring urban agriculture.
There is a little mistake in the paragraph 1.2, line132: the landscape images are from 1975 (and non 1979) to 2020.
Response:
The error has been corrected.

Round 2
Reviewer 2 Report
Comments and Suggestions for Authors
The suggestions to ameliorate the understanding of the methodology and to improve the conclusion have been taken into account. The methodology remains a bit complex but is understandable.